# Development of a Rapid Measurement Method for Analysis of the NOx Conversion Process Based on Quantum Cascade Laser Absorption Spectroscopy

**DOI:** 10.3390/s23083885

**Published:** 2023-04-11

**Authors:** Xi Yang, Zhirong Zhang, Shuang Yang, Pengshuai Sun, Bian Wu, Hua Xia, Runqing Yu

**Affiliations:** 1School of Environmental Science and Optoelectronic Technology, University of Science and Technology of China, Hefei 230026, China; 2Anhui Provincial Key Laboratory of Photonic Devices and Materials, Anhui Institute of Optics and Fine Mechanics, HFIPS, Chinese Academy of Sciences, Hefei 230031, China; 3Key Lab of Environmental Optics & Technology, Anhui Institute of Optics and Fine Mechanics, HFIPS, Chinese Academy of Sciences, Hefei 230031, China; 4Advanced Laser Technology Laboratory of Anhui Province, Hefei 230037, China

**Keywords:** nitrogen oxides, conversion process, chemical reaction of gas, laser spectroscopy, quantum cascade laser

## Abstract

In this study, a method for double-beam quantum cascade laser absorption spectroscopy (DB-QCLAS) was developed. Two mid-infrared distributed feedback quantum cascade laser beams were coupled in an optical cavity for the monitoring of NO and NO_2_ (NO at 5.26 μm; NO_2_ at 6.13 μm). Appropriate lines in the absorption spectra were selected, and the influence of common gases in the atmosphere, such as H_2_O and CO_2_, was avoided. By analyzing the spectral lines under different pressure conditions, the appropriate measurement pressure of 111 mbar was selected. Under this pressure, the interference between adjacent spectral lines could be effectively distinguished. The experimental results show that the standard deviations for NO and NO_2_ were 1.57 ppm and 2.67 ppm, respectively. Moreover, in order to improve the feasibility of this technology for detecting chemical reactions between NO and O_2_, the standard gases of NO and O_2_ were used to fill the cavity. A chemical reaction instantaneously began, and the concentrations of the two gases were immediately changed. Through this experiment, we hope to develop new ideas for the accurate and rapid analysis of the process of NOx conversion and to lay a foundation for a deeper understanding of the chemical changes in atmospheric environments.

## 1. Introduction

NO and NO_2_ are the primary pollutants in the atmosphere, and they mainly result from the burning of fossil fuels, such as during fuel combustion in thermal power plants, exhaust emissions from coal-fired boilers, and vehicle exhaust emissions [1,2,3,4]. Coal combustion exhaust gas is the most important source of NO_2_ in the atmosphere, and it is the focus of environmental monitoring. The organic nitrogen in coal fuels is easily decomposed and oxidized, leading to higher levels of NO_2_ during combustion, and NO is also produced during the combustion process [5]. With the increase in the NO concentration in the atmosphere, free radical reactions occur in the stratosphere and destroy the ozone [6,7]. In addition, nitric acid is generated at the bottom of the stratosphere and enters the clouds to form acid precipitation [8,9]. Moreover, it enters the troposphere after atmospheric transmission, and the tropospheric ozone concentration increases after photochemical reactions. Therefore, NO is also an important and easily neglected greenhouse gas [10,11]. NO_2_ is also one of the causes of acid rain, which leads to surface water acidification, eutrophication, and an increase in contents that are toxic for fish and other aquatic organisms [12,13]. These chemical processes can be described as follows: O_2_ in the upper atmosphere can become ozone (O_3_) after absorbing ultraviolet light, O_3_ can be decomposed (O_3_ → O_2_ + O), and O_3_ reacts with NO (NO + O_3_ → NO_2_ + 2O, NO_2_ + O → NO + O_2_) [14,15,16]. In order to control the emission of pollutants and deeply understand the actual process of their conversion, it is necessary to simultaneously and rapidly monitor the dynamic changes in NO and NO_2_ concentrations.

Non-dispersive infrared (NDIR) and differential optical absorption spectroscopy (DOAS) are common measurement methods for the simultaneous measurement of the two gases [17,18,19]. However, the process of measuring NO_2_ is highly dependent on the algorithm and is subject to many interfering factors. There are also different problems in the application of the direct measurement of NO_2_ in different industries. Laser absorption spectroscopy technology is based on the Lambert–Beer law, and it is a common method for the detection of trace gases [20,21,22]. With the continuous development of laser technology, the quantum cascade laser (QCL) has extended the measurement band of gas to the mid-infrared region, where there is a stronger absorption strength and more spectral lines [23,24,25,26,27]. Based on these advantages, quantum cascade laser absorption spectroscopy (QCLAS) can provide faster and more accurate monitoring capabilities, and it has become a research hotspot [28]. It has been applied in the fields of environmental monitoring, industrial processes, respiratory gases, and so on [29,30,31,32].

In this study, a gas detection system, based on the mid-infrared quantum cascade absorption spectrum, was developed for the simultaneous analysis of NO and NO_2_. By setting the pressure and different gas concentration ratios in the gas cell, the changes in NOx absorption spectra under different conditions were obtained through experimental analysis. The stability of the system was also analyzed. Finally, we simply analyzed the reaction process of NO and O_2_ to verify the feasibility of using QCLAS for realizing the detection of chemical reactions between gases.

## 2. Experimental System and Methods

### 2.1. Theory

For laser absorption spectroscopy, the absorption at frequency *v* is given by the Beer–Lambert law,
(1)$$I_{t}=I_{0}\exp\left[-S\left(T\right)CPL\varphi \left(\upsilon \right)\right],$$
where *I*_0_ and *I_t_* are the incident and transmitted laser intensity, respectively. *υ* [cm^−1^] is the wave number corresponding to the absorption line, *C* is the volume concentration fraction of gas, *L* [cm] is the optical path, *P* [atm] is the gas pressure, *T* [K] is the gas temperature during the measurement, *φ*(*υ*) is the absorption linear function normalized by the integral area, and *S*(*T*) [cm^−2^atm^−1^] is the line intensity of the absorption line when the gas temperature is *T*.

### 2.2. Experimental Setup

Figure 1 shows the QCLAS gas detection system. Two room-temperature QCL lasers were controlled by homemade low-noise mid-infrared laser controllers, and the output wavelengths were 5.26 μm for NO and 6.13 μm for NO_2_. The two collimated laser beams were coupled in the gas cell by using mirrors. The material of the gas cell was corundum, and both sides were sealed with CaF_2_ window pieces. The interior of the gas cell was cylindrical with a length of 70 cm and a diameter of 3 cm, and the volume of the gas cell was 490 mL. Then, the absorption spectra were focused on a thermoelectrically cooled mercury cadmium telluride photodetector. The two detectors were internally integrated with a multistage semiconductor refrigeration sheet (TEC) and a wedge-shaped zinc selenide window with antireflection film, which could effectively prevent optical interference. Finally, the spectral signals were acquired by using a data acquisition (DAQ) card and processed by a PC with software. The gas distribution system had two gas inlets and one gas outlet, which were connected to pipes. The system was able to control the gas flow and gas mixture ratio to achieve a good gas distribution effect. In addition, the cavity was connected to a mass flowmeter and pump, and the PID control method was adopted for high-precision pressure control. The pressure control circuit controlled the mass flowmeter and cooperated with the vacuum pump to achieve high-precision pressure control, with a control accuracy of ±0.01 mbar. Therefore, we were able to analyze the changes in NO and NO_2_ gas concentrations in the conversion process under different concentration ratios and pressures.

### 2.3. Transition Selection

For NO and NO_2_ molecules, there is a strong line strength in the mid infrared band. In this study, the transitions of NO at 5261.7233 nm, 5261.7235 nm, 5261.7237 nm, 5261.7254 nm, 5261.7256 nm, and 5261.7259 nm were selected based on the HITRAN database. However, because the wavelengths of these six groups were close and exceeded the resolution of the QCL bandwidth, they could be regarded as one group of absorption lines. Similarly, the transitions of NO_2_ at 6131.1097 nm and 6131.1232 nm were selected, and they were regarded as one group of absorption lines. Figure 2 shows the two gas absorption spectra. The data were derived from the HITRAN 2012 database and calculated for pressures of 1013 mbar (1 atm) and 101.3 mabr (0.1 atm), respectively, at room temperature and with an optical path length of 70 cm [33]. These spectral lines of the gases were selected to avoid the strong spectral lines of other gases, such as H_2_O and CO_2_, thus reducing the impact of cross-absorption. The relationship between the wavelength and driving current of the laser at the selected temperature is shown in Figure 2. This was measured by using a wavelength meter, and the wavelength range was able to cover the target wavelength. The drive currents of the lasers for NO and NO_2_ were 145 mA and 260 mA, respectively. The scanning frequency was 10 Hz.

## 3. Results and Discussion

Under normal conditions, the output wavelength of a QCL is determined according to the driving current, TEC temperature, and scanning voltage. At room temperature, the standard gas concentrations of 1032 ppm (NO) and 2000 ppm (NO_2_) were used to fill the cavity, and the different cavity pressures were set. Then, the direct absorption signals, shown in Figure 3a,b, were obtained. The decrease in pressure in the cavity affected the broadening of the spectral lines and reduced the cross-interference between the spectral lines. At the same time, the decrease in pressure also affected the intensity of the absorption signal. Therefore, according to the above analysis, we selected a stable pressure of 111 mbar for the following experiments.

Before each experiment, the cavity and pipeline were purged with pure nitrogen to eliminate the influence of interfering gas on the measurement results. In addition, various concentrations of NO_2_ and NO were configured with a high-performance gas distributor. During the entire process, the air pressure in the cavity was kept at 111 mbar. The average number of signals was eight. The direct absorption signals for NO and NO_2_ at different concentrations were measured, as shown in Figure 3c–f. By analyzing the relationships between the concentrations and intensities of the two groups of results, we were able to find that the fitting correlations were 0.993 and 0.998 for R^2^(NO) and R^2^(NO_2_), respectively. They had good linear relationships.

Meanwhile, the detection limits of the system were found by calculating the signal-to-noise ratio of the absorption signals. As shown in Figure 4a,b, the NO concentration was 236 ppm and the signal-to-noise ratio was 22.5, so the detection limit of 10.5 ppm was calculated. Similarly, the NO_2_ concentration was 167 ppm and the signal-to-noise ratio was 11.4, so the detection limit of 14.6 ppm was calculated. During the experiments, we recorded the relationship between the measured data and time, as shown in Figure 4c,d. We defined the response time of the system according to when the measured value reached 90% (T_90_) of the set value; this was T_90_ = 2.3 s for NO and T_90_ = 2 s for NO_2_.

For the purpose of analyzing the system’s stability, a mixed gas, consisting of both NO and NO_2_, was used to fill the absorption cell at a flow rate of 0.2 L/min; the concentration of NO was 545 ppm and that of NO_2_ was 770 ppm. The concentration data were continuously measured for 40 min, and the results are shown in Figure 5. From the relative histograms of each dataset, which are shown on the right side of Figure 5, it can be seen that the measurement results had a good Gaussian fit. Thus, the standard deviations of NO and NO_2_ were 1.57 ppm and 2.67 ppm, respectively. It is worth noting that the experiment described in this paper was intended to verify the feasibility of measuring the NOx conversion process. If we would like to achieve a higher accuracy and a lower measurement limit, a longer optical path or cavity ring-down spectroscopy (CRDS) and off-axis integrated cavity output spectroscopy (OA-ICOS) could be used based on the QCL technology [34,35,36].

The next experiment was carried out to verify that the system could monitor the changes in gas concentrations during a chemical reaction. The system was set up by filling it with NO gas at 1032 ppm: the gas flow rate was 0.2 L/min, the NO concentration was stable at 1032 ppm, and the concentration of NO_2_ was 0. After waiting for a period of time, O_2_ was used to fill the cell at the time indicated by the purple dotted line. It was found that when O_2_ was added, it immediately reacted with NO to produce NO_2_. NO was consumed in the reaction: its concentration decreased and the concentration of NO_2_ increased, as shown in Figure 6. Due to the slow gas flow, the limitation of the container’s volume and insufficient gas contact, the reaction speed slowed down after a period of time, and the reduction in the NO concentration and the increase in the NO_2_ concentration tended to become gentle. However, chemical reactions were still slowly occurring. Because the system maintained a dynamic state of intake and exhaust, NO and O_2_ remained present. Therefore, NO was not completely consumed in the reaction, and the concentration did not drop to 0. By analyzing the data, it was found that when O_2_ was used to fill the container, the chemical reaction occurred rapidly, and the measured concentrations of the two gases also changed immediately. This was sufficient to prove that the system could quickly monitor the changes in gas concentrations when the reaction began. Two tangent lines were obtained at the inflection points of the two concentration–time curves, as shown in Figure 6, where the NO concentration started to decrease and the NO_2_ concentration started to increase. The ratio of the slopes of the two tangent lines was 1.058. With reference to the chemical reaction equation (2NO + O_2_ → 2NO_2_), the rate at the beginning of the reaction had a certain correlation with the chemical formula, which was close to a 1:1 replacement. Therefore, the system was able to meet the requirements of simultaneous and rapid measurement of changes in gas concentrations during chemical reactions or the evolution of an atmospheric environment.

## 4. Conclusions

In this study, a dual-beam laser absorption spectroscopy gas detection system was developed based on two QCLs, and it could be used to monitor the changes in the concentrations of NO and NO_2_ in real time. The performance of the system was analyzed, and there was a good linear relationship between the absorption peaks and the concentrations. The experimental results showed that, for NO and NO_2_, the detection limits of the system were 10.5 ppm and 14.6 ppm, respectively. To assess the stability of the system, the concentrations of NO and NO_2_ were measured at the same time over 40 min, and the standard deviations of the concentrations were 1.57 ppm and 2.67 ppm. Therefore, this system is very helpful for monitoring the concentrations of NO and NO_2_, and it has positive significance for analyzing the conversion of NOx. In the future, we hope to continuously optimize this measurement system to achieve higher measurement accuracy and limits.

## Figures and Tables

**Figure 1 sensors-23-03885-f001:**
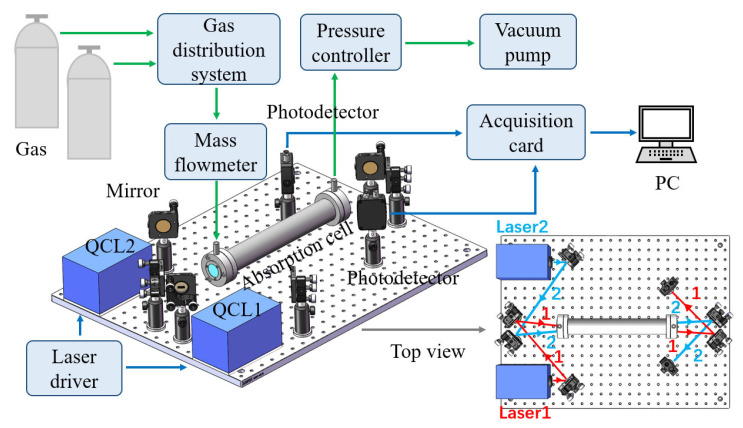
Schematic of the QCLAS gas detection system.

**Figure 2 sensors-23-03885-f002:**
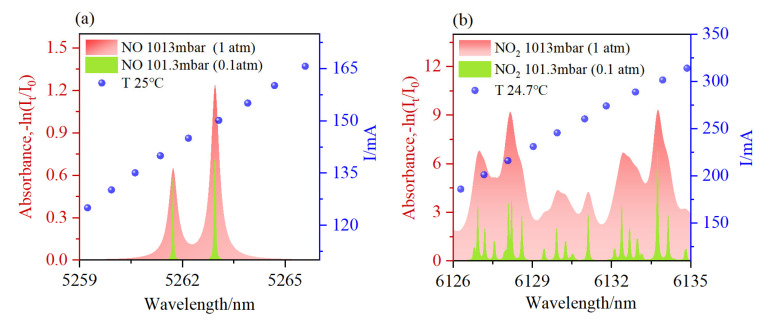
Absorption lines that were simulated based on the HITRAN database. (**a**) NO spectrum; (**b**) NO_2_ spectrum.

**Figure 3 sensors-23-03885-f003:**
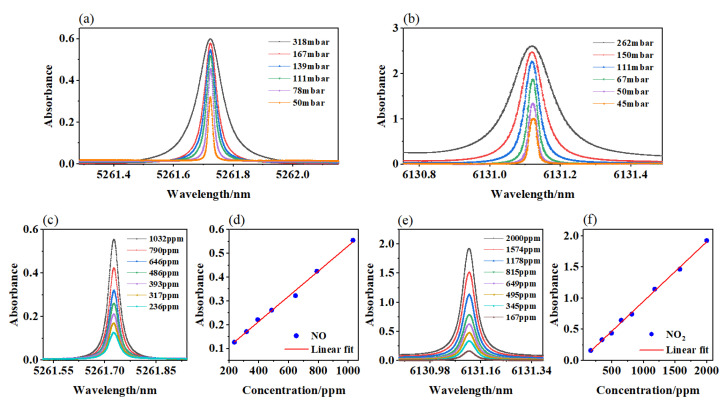
Direct absorption signals at different pressures: (**a**) NO signals; (**b**) NO_2_ signals. (**c**) The different direct absorption signals of NO. (**d**) Linear fit between the peak values and concentrations. (**e**) The different direct absorption signals of NO_2_. (**f**) Linear fit between the peak values and concentrations.

**Figure 4 sensors-23-03885-f004:**
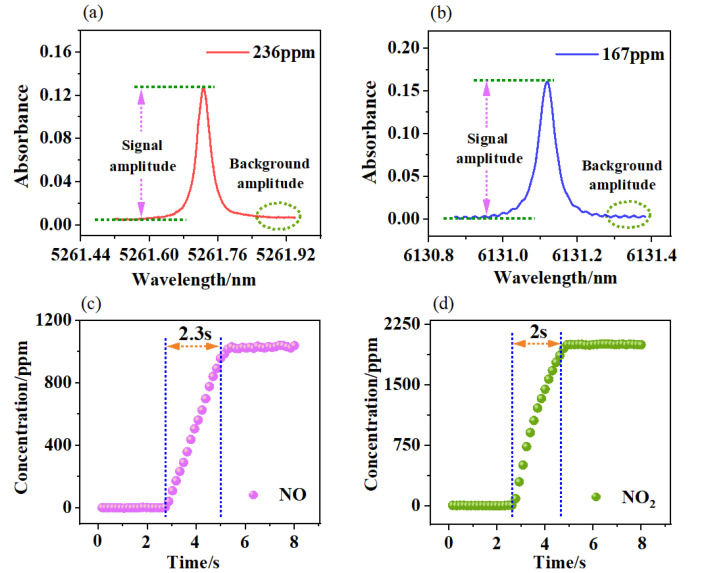
Direct absorption signals for (**a**) NO and (**b**) NO_2_. Response times for the QCLAS system with (**c**) NO and (**d**) NO_2_.

**Figure 5 sensors-23-03885-f005:**
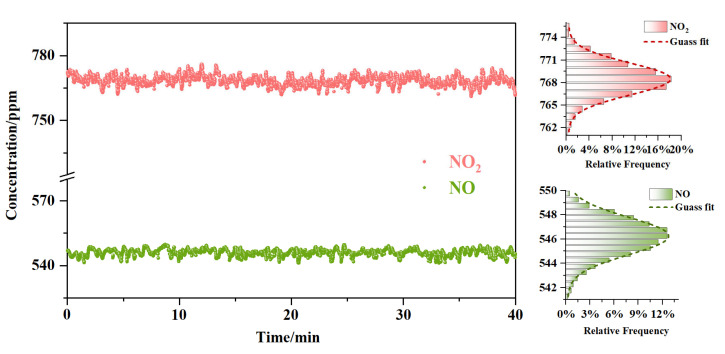
Continuous measurement of the concentrations of NO and NO_2_ for 40 min.

**Figure 6 sensors-23-03885-f006:**
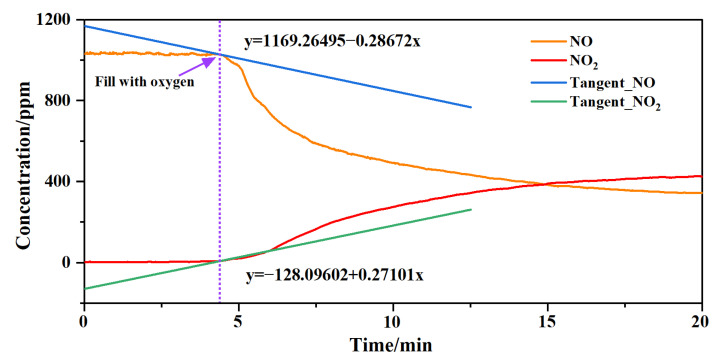
Changes in gas concentration when using O_2_ to fill an area containing NO.

## Data Availability

Not applicable.

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
