# Peer review of "Development of a Rapid Measurement Method for Analysis of the NOx Conversion Process Based on Quantum Cascade Laser Absorption Spectroscopy"

_sensors, 2023, doi:10.3390/s23083885_

Round 1

Reviewer 1 Report

In this paper, a dual-beam laser absorption spectroscopy gas detection system for NO and NO2 is developed based on two QCL lasers at 5.26 μm for NO and 6.13 μm for NO2, respectively. After optimizing gas sampling pressure, detailed experimental analysis was investigated, and the experimental results show that the standard deviations of 1.57 ppm and 2.67 ppm are obtained for NO and NO2, respectively. This paper can be recommended for publication. However, some details of experiments and techniques should be clarified for improving the manuscript.

1) The chemical reaction between NO and O2 was done for investigating chemical reactions or atmospheric environment evolution. Indeed, atmospheric NO and NO2 are generally at ppb level, but the concentration of NOx in industrial emissions is relatively high. for example, Environ. Monit. Assess. 68(1):37-50,2001;Metrology and Measurement Systems, XXII(1), 25-38, 2015; Science of The Total Environment 832: 154781,2022;Measurement 201 (2022) 111729. The related publicaitons of NOx in vehicular traffic, atmospheric background and heavily urbanized environment and coal-fred chemical plant can be discussed in Introduction, and then the authors can clarify their mission.

2) How long is the optical path of the spatial optical path outside the absorption cell in the system? Are there any environmental factors in the measurement process?

3) How about the solubility of NO2 and NO, whether they are soluble in water. Does water in the air affect the measurement results?

4) The experiment is carried out at room temperature. Does the temperature have any effect on the experiment? Whether the influence factor of temperature is considered. 

5) The last part of the experiment is to verify the feasibility of the system to detect the change of chemical reaction, the concentration of oxygen and the basis for selection. What is your basis? 

6) It is mentioned in the article that NO will not be completely consumed in the reaction. If the inlet and outlet are closed, will NO be completely consumed? Have you measured the result that NO concentration tends to zero. Please briefly analyze the reasons.

7) Finally, there are a few typos, such as

Line 66: the number of oxygen molecules should be in the subscript

Line 99-101: “5.26um and 6.13um” should be 5.26 μm for NO and 6.13 μm, please pay attention to the separator - space.

Line 217-218 References : capitalizing the initial letter

Reviewer 3 Report

In this paper, a system for measuring the concentrations of NO and NO2 is developed and is used for monitoring the chemical reaction between NO and O2. The paper is well organized while a few points still need to be addressed.

1) According to Eq.(1), the expression of absorbance in Fig.2 should be -ln(It/I0).

2) The meanings of the purple lines and blue dots in Fig.2 should be clearly defined.

3) It's recommended to draw the laser path 1 and 2 with different colors in Fig.1 in order to distinguish them well.

4) The formula 2NO+O22NO2 should always hold during the chemical reaction? If so, in Fig.6 the absolute value of the slope of the orange line should always equal that of the red one, and it's expected to plot the slopes versus time in a sub-figure.

Reviewer 4 Report

In this manuscript, a dual-beam laser absorption spectroscopy gas detection system is developed based on two QCL lasers, which can monitor the concentrations change process of NO and NO2 in real time. And developing a new idea for accurate and rapid analysis of NOX conversion process. The manuscript has a novel idea and clear logic, it can be published once the following comments are taken care of:

(1) As a place where the laser acts with two gases NO,NO2, the gas cell is described here too simply and its volume and physical dimensions should be described in detail.

(2) “The system can control the gas flow and gas mixture ratio, in order to achieve the good gas distribution effect. Also, the cavity is connected with the mass flowmeter and pump, and the PID control method is adopted for high-precision pressure control.”

In the control of gas flow and pressure control precision, there should be specific parameter descriptions.

(3) “As shown in Figure 4(a) and Figure 4(b), the NO concentration is 236 ppm and the signal-to-noise ratio is 22.5, so the detection limit can be calculated 10.5ppm·m. Similarly, the NO2 concentration is 167 ppm and the signal-to-noise ratio is 11.4, so the detection limit can be calculated 14.6ppm·m.”

The NO concentration described in the article is 236 ppm, and the detection limit of 10.5 ppm·m was obtained, and it should be explained here how the detection limit was obtained.

Reviewer 5 Report

The authors present an apparatus for the simultaneous detection of NO and NO2. They make use of two room temperature QCLs in a single pass geometry through a 70 cm gas cell. They discuss their selection of NO and NO2 transitions and ideal pressure for quantifying the concentration of both species. The find a detection limit 10.5 and 14.6 ppm for NO and NO2 respectively. They show the temporal response of both signals, stability (up to 40 minutes), and monitor the change of both concentrations from the reaction NO + NO2. 

The work is concise, and shows the strength of using the fundamental bands of molecules for trace gas sensing. Overall, their analysis appears to be correct. However, I find it does not show any appreciable improvements or is able to differentiate itself from other similar works. Other instruments which use two QCLs for monitoring  of NO and NO2 have reached 10 ppb limits with 1 s of averaging, a work which the authors failed to cite (DOI 10.1007/s00340-017-6742-7).  I therefore I recommend the work not be published in Sensors. I offer some feedback for strengthening the manuscript, which should be considered for future submissions.

Comments

1. There is no mention of detection bandwidth, averaging time, or sampling rate. These details help understand what the circumstances of the measurement  were and how to compare it with other works.

2. An Allan deviation analysis on the data in Fig. 5 would help interpret the stabililty of the system. 

3. Were the calibration plots from Figure 3 used to determine the concentrations in Figs 4 and 5? Or were the cross sections from HITRAN use? How does the HITRAN predicted spectrum compare with the measured values?

4. Why was any form of modulation avoided in the measurement? Such as an optical chopper? If the calibrations plots are used anyways, why not use wavelength modulation?

5. Apart from showing NO and NO2 can be simultaneously measured, a result which has already been demonstrated in the literature, what is the point of these measurements and result? Clear objectives would help assess if the new result is noteworthy. Is it for field or chamber work? And so on. 

Minor comments.

1. The analysis of the NO + O2 data is not straight forward in my opinion. For one, the tangent in Figure 6 does intercepts the NO2 data twice. Also, why not take the sum of the two curves to show that it adds to 1.0 at short time scales? A full model whould also be beneficial. 

2. The numerous grammar mistakes made it challenging to parse this paper. A significant overhaul would have to be made before resubmission. 

Round 2

Reviewer 2 Report

The article has responded to the issues I raised and made improvements accordingly. In my opinion, the revised manuscript can be considered for publication.

Author Response

Dear reviewer:

Thank you very much for your valuable suggestions, which have improved this article. Thank you very much for your reply, and wish you a happy life.

Reviewer 5 Report

The minor changes have added additional details regarding the cell geometry, pressure control, and scanning conditions. Otherwise, a few of the grammatical errors were fixed. There are still errors and omissions which need to be addressed. 

The authors responded saying that although this measurement is orders-of-magnitude less sensitive than other NO/NO2 monitoring experiments using QCLs, their applications have higher concentrations making this unnecessary.

1. They still omit the number of averages (or integration time) in their measurements within the text.

2. Section 2.3 starts with extremely incorrect spectroscopic information. NO2 has a bent geometry, not linear. NO cannot have an antisymmetric stretch since it is a diatomic. Please change this first sentence to be correct. Also, it is very odd to refer to a molecule's symmetry with the word "symmetrical". 

An explanation for the 2 s response time should be included. I would expect the system to be much faster. 

Minor

Still an extremely large number of grammatical errors and strange wordings, and I cannot go through this line by line to find them all. For example

52: Redundant "are common measurement methods for the simultaneous measurement "

155: Redundant "Meanwhile, the detection limits of the system can be calculated by calculating "

181: Missing subject: "Set up the system, fill with 1032ppm NO gas first, the gas flow rate is 0.2L/min,"

209, Missing subject: "Analyzed the performance of the system, it has a good linearity of the relationship between absorption peak and concentration."
